# A Review of Classical and Rising Approaches the Extraction and Utilization of Marine Collagen

**DOI:** 10.3390/biotech14020026

**Published:** 2025-04-03

**Authors:** Cesia Deyanira Gutierrez-Canul, Luis Alfonso Can-Herrera, Emmanuel de Jesús Ramírez-Rivera, Witoon Prinyawiwatkul, Enrique Sauri-Duch, Victor Manuel Moo-Huchin, Emanuel Hernández-Núñez

**Affiliations:** 1Tecnológico Nacional de México, Instituto Tecnológico Superior de Calkiní, Avenida Ah Canul S/N por Carretera Federal, Calkiní 24900, Campeche, Mexico; cdgutierrez@itescam.edu.mx; 2Tecnológico Nacional de México, Instituto Tecnológico Superior de Zongolica, km 4 Carretera a la Compañía S/N, Tepetitlanapa, Zongolica 95005, Veracruz, Mexico; ejramirezrivera@zongolica.tecnm.mx; 3School of Nutrition and Food Sciences, Louisiana State University Agricultural Center, Baton Rouge, LA 70803, USA; wprinya@lsu.edu; 4Tecnológico Nacional de México, Instituto Tecnológico de Mérida, km 5 Mérida-Progreso, Mérida 97120, Yucatán, Mexico; enrique.sd@merida.tecnm.mx (E.S.-D.); vmmoo@yahoo.com (V.M.M.-H.)

**Keywords:** fish material, acid-soluble collagen, ultrasound-assisted extraction, innovative collagen extraction methods

## Abstract

This comprehensive review explores the extraction and utilization of marine collagen, a sustainable alternative to traditional mammalian sources. The review covers conventional extraction methods like acid and pepsin solubilization, highlighting their limitations and contributing to the search for improved efficiency and sustainability. It also delves into innovative extraction technologies, such as ultrasound-assisted extraction, deep eutectic solvents, and supercritical carbon dioxide, showing their potential to revolutionize the field. The significance of collagen hydrolysis in generating bioactive peptides with diverse functionalities is also discussed, emphasizing their potential applications in various sectors. By providing an analysis of marine collagen extraction and its implications, this review presents a perspective for leveraging this valuable bioresource, promoting a circular economy, and satisfying the increasing demand for high-quality collagen in diverse industries.

## 1. Introduction

Collagen is well known for its crucial role in providing mechanical strength to the skin, bone, and cartilage of mammals [1]. Its unique triple-helical structure is composed of repetitive sequences of Gly-X-Y amino acids, with Gly at every third position, X being proline, and Y being hydroxyproline or lysine. Traditionally, mammalian sources have dominated collagen production; however, concerns regarding disease transmission (e.g., bovine spongiform encephalopathy, allergies, and avian influenza) and religious restrictions have encouraged the exploration of alternative sources, with marine organisms emerging as a compelling and sustainable reservoir [1,2]. Marine-derived collagen, particularly from fish skin, presents a multitude of advantages over its mammalian counterparts. It exhibits lower immunogenicity and aligns seamlessly with the vision of sustainable resource management by valorizing fish processing byproducts that would otherwise be relegated to waste streams [3,4].

The extraction of collagen from marine sources involves a series of steps comprising pretreatment, extraction, and recovery. The pretreatment stage encompasses the remotion of non-collagenous proteins using alkaline solutions, demineralization, and [5,6]. Following pretreatment, the extraction process aims to detach collagen from the tissue matrix (scales, skin, or another part of fish). Conventional methods, such as acid and pepsin solubilization, have been extensively employed; however, these techniques are often constrained by limitations such as low yields and prolonged processing durations [7]. The choice between acid-soluble collagen (ASC) and pepsin-soluble collagen (PSC) extraction lies in the specific application and desired collagen properties. ASC retains its native triple-helical structure, making it suitable for biomaterial development and tissue engineering, while PSC improves acid extraction by enhancing the solubility and bioavailability of fish collagen [8,9]. The quest for heightened efficiency and sustainability has fueled the exploration of innovative extraction technologies, with ultrasound-assisted extraction (UAE) emerging as a promising contender [10]. UAE is based on the application of high-frequency sound waves to produce cavitation, a phenomenon distinguished by the formation and subsequent implosive collapse of microbubbles within the extraction milieu. This process disrupts the fish material, facilitating solvent contact and increasing the mass transfer of collagen from the matrix to the liquid phase [11].

The present review delves into a comprehensive exploration of both classical and emergent techniques for extracting marine collagen, with a spotlight on the profound ramifications of UAE on extraction yield and the versatile properties of collagen. It will highlight the potential for UAE to effectively and sustainably transform fish-processing byproducts into valuable commodities, meeting the increasing demand for high-quality collagen in various industries. In addition, the review will explore the emerging field of novel extraction technologies, such as deep eutectic solvents, supercritical carbon dioxide extraction, and pulsed electric fields. These technologies can bring about a revolutionary era in marine collagen extraction, marked by increased efficiency, sustainability, and product quality. It is also important to recognize that the properties and applications of marine collagen can be further expanded through enzymatic hydrolysis, leading to the obtention of bioactive peptides with diverse functionalities [6,12]. The impact of hydrolysis conditions on the characteristics and bioactivities of these peptides will be elucidated, presenting their potential as functional food ingredients, cryoprotectants, and bioactive components in various sectors.

## 2. Extracting Fish Collagen

### 2.1. Pretreatment of Raw Fish Material

Before initiating protocols for obtaining collagen, the fish material must be cleaned. Firstly, the fish is rinsed with plenty of water; then, the tissue of interest is separated from the body. For soft tissue (e.g., skin, swim bladder), the particle size is usually reduced by cutting [13,14]. Hard tissue or mixed byproducts should be homogenized by grinding or trituration [15]. In the next step, the non-collagenous proteins are removed, usually by soaking the material in 0.1 M NaOH solution, although Ca(OH)_2_ can be used [16]. This process eliminates the telopeptide region, so replacing the NaOH solution from time to time for better effectiveness is essential. Thus, this step could take a few hours to several hours [17,18]. Collagen solubility in acid media is strongly affected by the telopeptide region [19]; in other words, the higher the remotion of the telopeptide region, the higher the collagen extraction. Finally, the fish material is defatted with 10% butyl alcohol for several hours [8]. Other authors include a demineralization step, commonly using ethylenediaminetetraacetic acid solution [5,7]. Besides the traditional protocol, some authors have included additional steps for improving collagen extraction. Vate et al., 2022 [9] used high-shear mechanical homogenization equipment to reduce the particle size of raw material. With the aid of this technology, the contact between the material and the solvents was promoted, so less acid and alkali solvents were required. Huang et al., 2016 [20] pretreated fish scales with an extrusion-hydro-extraction (EHE) process to facilitate collagen extraction. However, there is a concern about collagen denaturation because EHE requires temperatures below 25 °C [20].

### 2.2. Acid and Pepsin-Based Methods for Obtaining Fish Collagen

Acid media is the gold standard for collagen extraction, usually performed with acetic acid in no more than 0.5 M. ASC is obtained using an acetic acid solution, whereas PSC is obtained using an acetic acid solution supplemented with pepsin (0.1–1%). Zamorano-Apodaca et al., 2020 [15], obtained collagen from mixed products of fish species in a simple manner. Firstly, raw fish was treated with 0.5 M acetic acid at a 1:10 solid/liquid ratio for 24 h at 25 °C. The obtained collagen was Type I with >200 kDa of molecular weight and 68% of yield. Lino-Sánchez et al., 2023 [7] obtained Type I collagen from Biajaiba fish, a native species from Mexico. They were interested in ASC and PSC; thus, they combined the acetic acid and pepsin simultaneously or step by step in three different protocols. For methodology A, acid acetic containing 0.1% pepsin was used as extraction media. Methodology B consisted of two sequential extractions, the first using acetic acid containing 0.1% pepsin and the second using acetic acid containing 1% pepsin. Methodology C involved two extractions, the first with acetic acid only and the second with acetic acid containing 1% pepsin. The product obtained using each methodology was analyzed individually. Methodology A was the simplest protocol, and 0.51% collagen was obtained. By the first step of methodology B (like methodology A), a 1.11% yield was obtained. This difference was probably because of the variability among scale batches. In the second step of methodology B, a 0.74% yield was achieved. However, it is important to note that 1% pepsin was used. For methodology C, 0.57% and 1.11% yields were obtained. In this case, the raw material was treated with acetic acid first, then 1% pepsin was used. The results showed that the scales of Biajiba fish have a low collagen content. Higher yields were obtained after the second extraction cycle, but higher quantities of pepsin were required. Cruz-López et al., 2021 [14], chose the skin and swim bladder of Gulf Corvina, as both organs are rich in collagen. In this case, the yield of skin collagen was 82% while the yield of swim bladder was 69%, the latter richer in hydroxyproline than the former. The degree of hydroxylation of proline was 30.4% for the skin and 33.5% for the swim bladder, suggesting that collagen from the swim bladder had the highest thermal stability between the two tissues. This hypothesis was confirmed by DSC analysis; the denaturation temperatures of the skin and swim bladder were 29.8 °C and 32.5 °C, respectively. The higher denaturation temperature and hydroxyproline content demonstrated the superior quality of swim bladder collagen. In this sense, García-Sifuentes et al., 2021 [21], obtained ASC and PSC from discarded fish products. The ASC contained 7% of hydroxyproline, while the PSC contained 6%. The denaturation temperature was 38.27 °C and 38.07 °C for ASC and PSC, respectively. Although the extraction based on acetic acid showed a higher yield than the pepsin-based protocol (4.4% vs. 2.2%), the quality of the collagen was comparable, considering the hydroxyproline content and denaturation temperature were similar. This could be related to the susceptibility of native collagen to extraction depending on pH, fish species, and fish organs. For example, Medina-Medrano et al., 2019 [22], studied fractions extracted from red tilapia skin and gills. It was possible to extract collagen from the skin by saltine, acidic, and pepsin media (1.46%, 3.02%, and 2.52% yields, respectively). In contrast, collagen extraction from gills was only possible with pepsin (0.16% yield). Arumugam et al., 2018 [23] applied the Box–Behnken statistical analysis to optimize the extraction of sole fish skin collagen. In their methodology, the fish skin was treated with acetic acid and subsequently with NaCl. The study revealed that the maximum yield (19.27 mg of collagen per gram of fish skin) was obtained by 0.54 M acetic acid, 1.9 M NaCl, 8.9 mL of solvent per gram of solid, and 32 h of extraction time. Acetic acid is the most popular method for obtaining collagen extract; however, the yields are low, and extraction times are very long. High yields are achieved only in cases where the fish organ is known to have collagen as the predominant component. Bhuimbar et al., 2019 [24], explored the efficiency of organic and inorganic acids to obtain collagen from black ruff. They carried out the extraction with acetic acid, citric acid, lactic acid, formic acid, tartaric acid, sulfuric acid, and hydrochloric acid. Lactic acid showed the best performance among organic acids, with a 45% collagen yield (wet basis) after 72 h of extraction. Under similar conditions, only 25% of collagen was recovered by acetic acid. The hydroxyproline content in collagen extracted by lactic and acetic acid was 4.83 and 2.91 mg/g, respectively. This study opens the possibility of exploring alternative organic acids for collagen extraction. In contrast, sulfuric acid and hydrochloric acid were inefficient in extracting collagen [10], probably because they are extremely corrosive. Table 1 and Table 2 show comparative data about fish collagen yields obtained by different protocols. Figure 1 shows a general diagram of the classical methods and rising approaches for collagen extraction in fish.

### 2.3. Ultrasound-Assisted Extraction

For humans, the capacity of the hearing sense is between 20 kHz and 20 kHz; ultrasound is below that limit. In liquid media, ultrasound produces cavitation through the formation, growth, and collapse of bubbles [27]. This process is violent and generates high pressure and temperature locally. In fish material extraction, ultrasound disrupts the intermolecular bonds that promote the release of compounds of interest [17]. Scientists have taken advantage of US technology to improve acid- and pepsin-based collagen extraction yields (Table 1). Heidari and Rezaei, 2022 [25], obtained collagen from tuna yellowfin skin and compared traditional pepsin extraction with and without ultrasound assistance. For the first case, the raw material was immersed in 0.5 M acetic acid containing 1% pepsin for 24 h. Other samples were immersed in the same extraction media but were assisted for 5, 10, 15, and 20 min of ultrasound. The collagen yield of samples treated with pepsin was 18.5%, whereas the extraction assisted by 15 min of ultrasound increased the collagen yield to 23.84%. Longer times resulted in the structural damage of pepsin, reducing its effectiveness, while less time was not enough to hydrolyze the protein. Moreover, higher purification was accomplished thanks to ultrasound assistance, and this was revealed because the highest content of proline, hydroxyproline, and glycine was found in samples treated during 15 min of ultrasound. The most important feature of ultrasound technology is that it requires only a few minutes compared to several hours, which are usually necessary to perform acid extraction. In addition, higher yields are achieved without sacrificing collagen quality (regarding amino acid content) [17]. Interestingly, ultrasound technology could efficiently remove the telopeptide region, producing collagen with higher thermal stability [13]. Song et al., 2018 [26] scaled up the ultrasound-assisted collagen extraction of flatfish skin from the laboratory to an industrial reactor. Firstly, they assessed the method using an ultrasound extraction system with 8 L of capacity and obtained 30.3% and 40.2% protein yields after 1.5 and 3 h of ultrasound treatment, respectively [28]. SDS-PAGE confirmed the correct extraction of native collagen, as the b, a1, and a3 bands were observed. After that, the process was replicated in an industrial extraction system of 60 L, and comparative yields to the laboratory system were achieved (31.3 and 46.2% of protein yield for 1.5 and 3 h of extraction, respectively). Moreover, the quantification of proteins of the collagen triple helix was 24.5, 12.7, and 18.8% for Gly, Hyd, and Pro. Ultrasound technology offers an important advantage regarding extraction times, and it is clear that a few minutes of sonication are required to obtain collagen with higher yields and similar physicochemical properties compared to acid extraction. However, it is important to emphasize that ultrasound generates hot temperatures locally, so developing the extraction under freezing conditions is critical to avoid collagen denaturation. Defining operational conditions (e.g., frequency, power, amplitude, and time) ensures that the extracted collagen maintains the highest quality without undesirable low-molecular fractions [11]. Ultrasonic treatment is a widely adopted method for collagen extraction due to its efficiency and effectiveness. The quality of collagen extraction is significantly influenced by several parameters, including frequency, power, amplitude, and time. The frequency of ultrasonic waves determines the energy delivered to the collagen matrix. Higher frequencies (typically above 20 kHz) can enhance the extraction process by breaking down collagen fibers more effectively. However, excessively high frequencies may lead to collagen degradation, reducing its quality [29]. The power of ultrasonic waves is directly related to the intensity of the treatment. Higher power levels can increase the efficiency of collagen extraction by providing more energy to disrupt the collagen matrix. However, excessive power can cause overheating and the potential denaturation of collagen, negatively affecting its quality [30]. Amplitude refers to the displacement of the ultrasonic probe. Higher amplitudes can enhance the cavitation effect, leading to more efficient collagen extraction. However, like power, excessively high amplitudes can damage the collagen structure, resulting in lower quality collagen [31].

### 2.4. Arising Technologies for Fish Collagen Extraction

Without leaving traditional technologies to obtain high-value-added products from discarded fish materials, other technologies have been developed to reduce processing time and increase the quality of fish collagen. Deep eutectic solvents (DESs) are a complex system formed by a hydrogen-bond-acceptor solvent and a hydrogen-bond-donator solvent; the mixture has a low melting point compared to the forming components [32]. Usually, a DES is used for the extraction of biomolecules easily affected by organic solvents or for those with low solubility in alcohol solvents [33]. Thinking about reducing risks of collagen denaturation and augmenting the quality of extracted collagen, Bisht et al., 2021 [34], screened several DESs to find out that urea–lactic acid and urea–propionic acid offered a higher fish collagen Type 1 yield than acetic acid. In addition, the SDS-PAGE showed that collagen extracted by DESs was highly pure, as no traces of low-molecular-weight species were detected. Following this method, Silva et al., 2024 [35], obtained high-quality collagen from codfish skin. Additionally, Bai et al., 2017 [36], reported an extraction efficiency of 90% for collagen from cod skin using a deep eutectic solvent (DES) composed of choline chloride (CC) and oxalic acid (OA). They also examined the impact of temperature on the extraction yield with the CC:OA mixture.

Supercritical carbon dioxide is a well-known technique for decaffeination and obtention of essential oils. The process involves submerging the samples in water acidified with CO_2_ as extraction media; then, the system is subjected to 31 °C and 50 bar to let the media reach the supercritical point. With this method, Sousa et al., 2020 [3] obtained a remarkable 13% yield of fish collagen from Atlantic cod fish skin with Pro and hydroxyproline content higher than that of the collagen obtained by acid-based extraction. Pulsed electric fields (PEFs) are a non-thermal technology used for food preservation based on applying high-voltage electric pulses (10 to 80 kV) for short periods. This process causes the electroporation of the cell membranes of organisms responsible for food spoilage. Its main advantage is that the food preserves its flavor, color, texture, and, more importantly, its nutritional properties [37]. He et al., 2019 [38], studied the extraction of collagen from the fishbone of Aristichthys nobilis by supplementing the liquid extraction media with trypsin, papain, or pepsin. Among the enzymes studied, 1% pepsin at a 1:10 ratio (raw material: liquid) yielded 13% mg/mL collagen under a 20 kV/cm electric field and eight pulses. Pulsed electric fields (PEFs) facilitate collagen remodeling by temporarily reducing collagen production and enhancing its degradation through the activation of matrix metalloproteinases (MMPs), influenced by reactive oxygen species (ROS). Additionally, PEFs down-regulate transforming growth factor beta 1 (TGF-β1), a key regulator of fibrosis, and significantly reduce the expression of crucial enzymes involved in extracellular matrix cross-linking. These effects suggest a promising therapeutic application for PEFs in the treatment of fibrosis [39]. Lin et al., 2020 [40], demonstrated that a single-pulsed electromagnetic field (SPEMF) at 0.2 Hz up-regulated tenogenic gene expression (Col1a1, Col3a1, Scx, Dcn) and down-regulated inflammatory gene expression (Mmp1) in vitro. After five days of SPEMF stimulation (3 min per day), there was a significant increase in collagen type I and total collagen synthesis protein levels. Additionally, under pro-inflammatory cytokine (IL-1β) irritation, the decreased expression of Col1a1 and Col3a1 was up-regulated by SPEMF treatment, and the increased expression of Mmp1 was reversed.

These emerging technologies are still under exploration. More studies applying DESs, supercritical CO_2_, and PEFs are necessary to determine their advantages and disadvantages, considering costs, time consumption, and life-cycle analysis.

Fish collagen is a rich source of amino acids, including glycine, proline, and hydroxyproline. These amino acids are crucial for the structural integrity and function of collagen. Glycine and proline are collagen’s most abundant amino acids, making up about one-third of its total amino acid content. It plays a vital role in maintaining the stability and structure of collagen’s triple-helix formation. On the other hand, hydroxyproline is a unique amino acid found exclusively in collagen. The post-translational modification of proline forms it and is crucial for the stability of the collagen triple helix. The content of hydroxyproline in fish collagen can vary, but it typically accounts for around 6–7% of the total amino acids [21]. It can be shown that there are variations in the concentrations of the amino acids in Table 2.

### 2.5. Application of Various Forms of Collagen Extraction

Collagen can be extracted in two forms: non-hydrolysate or hydrolysate, each one with unique features. The former preserves its triple helix configuration, whereas the second one is a fragmented form of the macromolecule, and each peptide has a different amino acid sequence. The key difference lies in the degree of hydrolysis: non-hydrolysate involves minimal hydrolysis, has larger collagen molecules, and retains its native structure, and hydrolysate involves extensive hydrolysis and has smaller peptides, with enhanced bioavailability, and bioactivity. The choice of extraction method depends on the intended application of the collagen. For example, acid soluble collagen (ASC) might be preferred for biomedical applications requiring structural integrity, while PSC hydrolysates are favored for their functional properties in health and beauty products [41].

In native tissues, collagen configuration confers the mechanical resistance of skin and tendons because of the aligned fibrillar arrangement [42]. As non-hydrolyzed collagen has a high molecular weight and triple helix conformation, it is applied to biomaterial development. For example, Jin et al., 2019 [43] used fish collagen to reinforce poly(lactide-co-glycolide) fibers. As expected for a structural protein, the fish collagen improved the tensile strength and elastic modulus of fibrous mats. In addition, the fibrous mat supported the bone mesenchymal stem cells, making it a promising scaffold for bone tissue engineering. In contrast, Zhu et al., 2024 [44] evaluated the degree of collagen denaturation in relation to the inhibition of cancer cell migration. Depending on the denaturation level, collagen changes its physicochemical properties that interfere with cellular functions, avoiding the growth of malignant cells. Another relevant aspect of this study is the replacement of mammalian collagen with fish collagen, which reduces the risk of transmission of diseases.

After hydrolysis, collagen breaks down into peptides, losing its structural features. The human body easily absorbs these small fragments. For this reason, hydrolyzed collagen has a high added value in the food industry for dietary supplements. Collagens are enzymes hydrolyzed. Pepsin is the most used enzyme for collagen extraction from fish material because it breaks the telopeptide region [6]. Hema et al., 2017 [45], compared the performance of pepsin, papain, and protease to hydrolyze collagen from Malabar grouper skin. They found that pepsin had the lowest yield (10% hydrolysis degree) among all enzymes. Thus, other proteolytic enzymes are preferred for collagen peptide obtention. According to previous reports, hydrolysis degrees of fish collagen below 20% have been achieved by using the enzyme alcalase [46,47]. To avoid the disadvantages of the enzymatic approach, such as long-lasting treatment, the use of organic solvents, and the difficulties in stabilizing pH, Ahmed and Chun 2018 [48] used subcritical water to obtain peptides from bigeye tuna skin fish. Subcritical water extraction (SWE) uses water at high temperatures (100–374 °C) and pressures to maintain it in a liquid state. This method is environmentally friendly as it avoids the use of organic solvents (green technology). SWE can break down proteins into peptides more rapidly than enzymatic hydrolysis. The high-temperature and -pressure conditions accelerate the hydrolysis process (Efficiency). SWE often results in higher yields of peptides compared to enzymatic hydrolysis due to the efficient breakdown of proteins [49]. This method involves heating the water below its critical point (374 °C and 218 atm). At this point, the water maintains a liquid state, but its physicochemical properties, such as density, viscosity, and dielectric constant, have changed. Subcritical water shows better solubility of organic compounds compared to water at room temperature and pressure.

Collagen peptides, obtained through various methods, have several applications, particularly in the food industry. Medina-Medrano et al., 2019 [22], explored the antioxidant capacity of collagen in its non-hydrolyzed and hydrolyzed forms. They found that hydrolyzed fractions, obtained through pepsin treatment, trapped free radicals more effectively than non-hydrolyzed ones due to their lower molecular weight. Zamorano-Apodaca et al., 2020 [15], obtained hydrolyzed collagen from mixed byproducts of fish species. Initially, raw fish was treated with 0.5 M acetic acid at a 1:10 solid/liquid ratio for 24 h at 25 °C. The hydrolyzed fractions were then obtained through enzymatic treatment with alcalase at pH 8. The resulting collagen was Type I, with a molecular weight greater than 200 kDa, a yield of 68%, and 5.46% hydrolysis achieved. After enzymatic treatment, five fractions with molecular weights ranging from 1 to 27 kDa were identified. These fractions exhibited radical scavenging activity for DPPH and OH radicals, comparable to standard Vitamin C. Moreover, the lower-molecular-weight fractions were richer in soluble proteins, making them suitable for functional foods. In another study, Hernández-Ruiz et al., 2023 [50], tested non-hydrolysate and hydrolysate collagen against E. coli. They identified two hydrolyzed fractions: F1 with peptides of 5–10 kDa and F2 with peptides of molecular weights lower than 5 kDa. The authors found that the antibacterial activity of collagen peptides was higher than that of non-hydrolysate collagen, likely due to the methionine content, especially in F1. Fimbres-Romero et al., 2021 [51] investigated the ability of hydrolyzed collagen extracted from jumbo squid and Sierra fish to inhibit ABTS radicals. They used enzymatic extracts obtained from Sierra fish viscera and jumbo squid hepatopancreas. Their findings revealed that low-molecular-weight peptides (<3 kDa) showed the best antioxidant performance compared to fractions of 3–10 kDa and >10 kDa. Additionally, Chel-Guerrero et al., 2020 [52], explored the potential of hydrolyzed collagen from Lionfish muscle as chelating agents for Cu^2+^ and Fe^2+^. In this study, the hydrolysate was obtained via two routes: the first using pepsin alone and the second using pepsin followed by pancreatin. In this study, the authors obtained hydrolysate with two routes, the first using pepsin and the second using pepsin followed by pancreatin. The findings revealed that hydrolyzed collagen obtained using the pepsin/pancreatin system exhibited higher metal chelation activity compared to collagen hydrolyzed with pepsin alone. This enhanced activity was attributed to the higher levels of arginine (Arg) and proline (Pro) obtained through the second enzymatic system. The content of imino and amino acids in collagen hydrolysates determines other functionalities. For instance, Heidari and Rezaei, 2022 [25], obtained hydrolyzed collagen from yellowfin tuna using ultrasonication for different durations (0–20 min). The samples treated for 15 min showed the highest proline and hydroxyproline content. This increase in proline and hydroxyproline content positively impacted emulsifying and foaming properties, making them useful for food and beverage applications [17,25]. For Cao et al., 2023 [12], those amino acids of a hydrophilic nature were important for developing cryoprotectants. Frozen food is supplemented with a cryoprotectant, an additive that protects food from freezing and thawing damage. Cryoprotectants help preserve the texture, flavor, and overall quality of frozen foods by preventing the formation of large ice crystals and maintaining cellular structure. They found a peptide fraction (454 Da) with an antifreeze effect like phosphorous, the commercial antifreeze agent most used.

## 3. Conclusions

Fish collagen has enormous potential to replace mammalian collagen without health concerns (digestive issues, allergic reactions, unpleasant taste, and regulation and efficacy). Fish byproducts are widely available around the world, making raw material almost cost-free. The wide application of high-molecular-weight collagen or collagen in hydrolysate form offers unlimited potential uses in the food, medical, cosmetic, and nutraceutical industries. Revalorizing fish byproducts is crucial for promoting a circular economy, reducing waste, improving resource efficiency, and creating high-value products. This supports environmental sustainability by reducing waste disposal impact and promoting sustainable fishing practices.

## Figures and Tables

**Figure 1 biotech-14-00026-f001:**
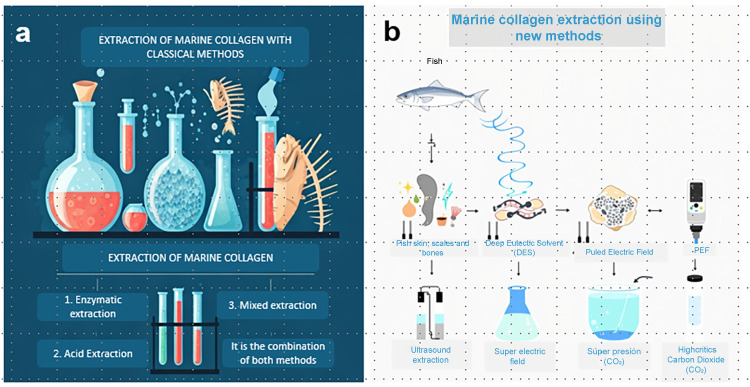
Graphic representation of collagen extraction methods. (**a**) Extraction of marine collagen whit classical methods. (**b**) Extraction of marine collagen using new methods.

**Table 1 biotech-14-00026-t001:** The collagen yield of different parts of different species under different treatment conditions.

Specie	Habitat	Tissue	Collagen Type	Extraction Method	Yield (%)	Ref
Mixed byproducts	Mexico	Skin Heads Skeleton	I	0.5 M acetic acid 1:10 *w*/*v* 24 h 25 °C	68.39% protein content	[15]
*Lutjanus synagris*	Gulf of Mexico	Scales	I	0.1% Pepsin 0.5 M Acetic acid 1:10 *w*/*v* 72 h	1% collagen dry weight	[7]
*Cynoscion othonopterus*	Gulf of Mexico	Skin Swim bladder	I	2% *w*/*w* Pepsin in 0.5 M Acetic acid 1:40 *w*/*v* 24 h 4 °C	82% (skin) 69% (swim bladder) Collagen Dry weight	[14]
Discarded fish products	Mexico	Leftovers	I	0.5 M Acetic acid 0.01% Pepsin 24 h	4.4% (ASC) 2.2% (PSC) Wet weight	[21]
*Oreochromis* sp.	Mexico	Skin Gills	I	Sequentially 0.05 M Tris, pH 7.2 0.5 M Acetic acid 1% *w*/*w* pepsin	1.46% 3.02% 2.52% Collagen Dry weight	[22]
Sole fish	India	Skin	I	0.54 M acetic acid 1.9 M NaCl 8.97 mL/g solvent/solid 32.32 h	19.27 mg/g protein content	[23]
*Centrolophus niger*	India	Skin	I	0.5 M lactic acid 0.5 M acetic acid 4 °C 72 h	45% wet basis 25%	[24]
*Thunnus albacares*	Iran	Fin skin	I	0.05 M acetic acid 1:30 *w*/*v* US 20 kHz, 300 W 25 min 3–5 °C	57% protein dry weight	[17]
*Thunnus albacares*	Iran	Fin skin	I	1% pepsin extract 0.5 M acetic acid 1:30 *w*/*v* US 400 W 15 min	23% protein dry weight	[25]
*Paralichthys olivaceus*	Korea	Skin	I	0.05 M acetic acid US 20 kHz 4 °C	31.3% (1.5 h) 46.2% (3 h) Protein dry weight	[26]

**Table 2 biotech-14-00026-t002:** Glycine, proline, and hydroxyproline content of fish collagen extracted by different methods.

	ASC					PSC					Ref
Source	Gly	Pro	Hyp	Td (°C)	dH (J/g)	Gly	Pro	Hyp	Td (°C)	dH (J/g)	
Mixed byproducts	180.5 mg/g	82.7 mg/g	43.9 mg/g								[15]
*Cynoscion othonopterus* skin						316 *	117 *	51 *	29.86	0.3	[14]
*Cynoscion othonopterus* Swim bladder						303 *	106 *	81 *	32.5	0.45	[14]
Discarded fish products	27%	15%	7%	38.27	0.64	26%	9%	6%	38.07	0.33	[21]
*Centrolophus niger* (lactic acid)			4.83 mg/g								[24]
*Centrolophus niger* (lactic acid)			2.91 mg/g								[24]
*Thunnus albacares*	333 *	112 *	70.2 *								[25]
*Thunnus albacares*	264.5 *	96.7 *	101.7 *	28.2							[17]
*Paralichthys olivaceus*	24.51%	18.8%	12.7%								[26]

* Residues/1000 amino acids.

## Data Availability

No new data were created or analyzed in this study.

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
