# Peer review of "A Review of Classical and Rising Approaches the Extraction and Utilization of Marine Collagen"

_biotech, 2025, doi:10.3390/biotech14020026_

Round 1

Reviewer 1 Report

Comments and Suggestions for Authors

No comment.

Author Response

Dear reviewer, thank you very much for your support of the manuscript.

Reviewer 2 Report

Comments and Suggestions for Authors

The authors have provided a review of fish collagen extraction procedures. While the information within the paper is valid and well put-together, it lacks comprehensiveness.

For example, only acid/pepsin/ultrasound based methods are discussed in greater detail. Would recommend adding more detail on DES, PEF, and supercritical CO2. In addition, other technologies (e.g., heat, salt extraction) should also be explored.

Secondly, just as the authors have discussed a Pretreatment step, the authors are recommended to add commentary on additional steps that follow extraction, i.e., purification and concentration steps which are also equally critical to generate a high quality product. Please add techniques used for these steps in your review to be truly comprehensive.

Thank you!

Author Response

Revisor 2.

The authors have provided a review of fish collagen extraction procedures. While the information within the paper is valid and well put-together, it lacks comprehensiveness.

For example, only acid/pepsin/ultrasound based methods are discussed in greater detail. Would recommend adding more detail on DES, PEF, and supercritical CO2. In addition, other technologies (e.g., heat, salt extraction) should also be explored.

In section 2.3. Ultrasound-assisted extraction, the use of ultrasound in collagen extraction is explained in detail.

Pulsed Electric Fields (PEFs) facilitate collagen remodeling by temporarily reducing collagen production and enhancing its degradation through the activation of matrix metalloproteinases (MMPs), influenced by reactive oxygen species (ROS). Additionally, PEFs downregulate transforming growth factor beta 1 (TGF-β1), a key regulator of fibrosis, and significantly reduce the expression of crucial enzymes involved in extracellular matrix cross-linking. These effects suggest a promising therapeutic application for PEFs in the treatment of fibrosis (Gouarderes et al. 2022).

(Lin et al. 2020), demonstrate that a single pulsed electromagnetic field (SPEMF) at 0.2 Hz up-regulated tenogenic gene expression (Col1a1, Col3a1, Scx, Dcn) and down-regulated inflammatory gene expression (Mmp1) in vitro. After five days of SPEMF stimulation (3 minutes per day), there was a significant increase in collagen type I and total collagen synthesis protein levels. Additionally, under pro-inflammatory cytokine (IL-1β) irritation, the decreased expression of Col1a1 and Col3a1 was up-regulated by SPEMF treatment, and the increased expression of Mmp1 was reversed.

Additionally, (Bai et al. 2017) reported an extraction efficiency of 90% for collagen from cod skin using a deep eutectic solvent (DES) composed of choline chloride (CC) and oxalic acid (OA). They also examined the impact of temperature on the extraction yield with the CC:OA mixture.

Secondly, just as the authors have discussed a Pretreatment step, the authors are recommended to add commentary on additional steps that follow extraction, i.e., purification and concentration steps which are also equally critical to generate a high-quality product. Please add techniques used for these steps in your review to be truly comprehensive.

The idea of ​​the review article is to expose the different extraction methods, the reviewer's suggestion is very important, but it is not in the interest of the authors.

Thank you!

REFERENCES

Bai C, Wei Q, Ren X (2017) Selective Extraction of Collagen Peptides with High Purity from Cod Skins by Deep Eutectic Solvents. ACS Sustain Chem Eng 5:7220–7227. https://doi.org/10.1021/ACSSUSCHEMENG.7B01439/SUPPL_FILE/SC7B01439_SI_001.PDF

Gouarderes S, Ober C, Doumard L, et al (2022) Pulsed Electric Fields Induce Extracellular Matrix Remodeling through Matrix Metalloproteinases Activation and Decreased Collagen Production. J Invest Dermatol 142:1326-1337.e9. https://doi.org/10.1016/J.JID.2021.09.025

Lin CC, Wu PT, Chang CW, et al (2020) A single-pulsed electromagnetic field enhances collagen synthesis in tendon cells. Med Eng Phys 77:130–136. https://doi.org/10.1016/j.medengphy.2019.12.001

Reviewer 3 Report

Comments and Suggestions for Authors

In this study, marine-derived collagen and mammalian-derived collagen are compared, and the efficiency and physical and chemical properties of collagen extracted by traditional and new methods are discussed, and the application of collagen is finally expanded. However, there are some several scientific concerns about this manuscript.

  1. Lines18-19 and 24-25, These contents are not reflected in the title, and the title is limited.
  2. Lines39 and 43-45, "Traditionally, mammalian 38 sources have dominated collagen production" and “Marine-derived collagen, particularly from fish skin, presents a multitude of advantages over its mammalian counterparts. It exhibits lower immunogenicity” lack reference support.
  3. Lines111-112, Please determine whether the decimal places such as 0.51% and 1.11% are correct.
  4. Lines134-140, Please determine the significance of this reference? This paragraph mainly discusses the traditional acquisition of collagen.
  5. Lines159-161, Please unify the expression of collagen yield. In the article, “45% collagen yield (wet basis)”, “25% of collagen” and “83 and 2.91 mg/g” cannot be compared, and the conclusion cannot be convincing.
  6. Table 1, The content shown in Table 1 is only the collagen yield of different parts of different species under different treatment conditions, so the title cannot be expressed as "Yields of acid-soluble collagen and pepsin-soluble collagen with and without ultrasound assistance".
  7. Lines 174-175, “Scientists have taken advantage of US technology to improve acid and pepsin-based collagen extraction yields (Table 1)". This sentence is recommended to delete because the contents in the table are already displayed below.
  8. Lines 191-199, This part should better illustrate that ultrasound technology can produce collagen with better thermal stability. Reference recommendations are replaced.
  9. Lines 204-206, It is recommended to discuss the impact of frequency, power, amplitude, and time during ultrasonic treatment on collagen extraction quality, or give examples.
  10. Lines 216-219, It is recommended to supplement the collagen yield under this method, and summarize and compare the several new collagen extraction methods and collagen yields introduced in the article.
  11. Table 2, The article does not show the role of glycine, proline and hydroxyproline in the extraction of collagen, and the significance of exploring the content of these three amino acids should be supplemented.
  12. Line244, No paragraph above mentioned the relevant methods or concepts for hydrolysis and non-hydrolysis of collagen. This paragraph does not connect with the previous article and is recommended to rewrite it.
  13. Line 245,Please explain the relationship between the extraction method mentioned above and the hydrolysate and the non-hydrolysate.
  14. Lines 249-250, Please add correct references.
  15. Line 265,Please explain what "enzymatic pools" is, and no application examples of enzymatic pools are found below.
  16. Lines 275-277, This passage does not explain the advantages of using subcritical water to extract peptides compared to enzymatic hydrolysis.
  17. Lines 281-303, The content is confusing, and it is recommended to reorganize this part of the content, for example, it is recommended to merge antioxidant abilities (DPPH, OH-, ABTS radicals).
  18. Lines 306-312, This paragraph is not an explanation of the use of hydrolyzed collagen as a chelating agents studied by Chel-Guerrero et al., and it is recommended that this part be discussed separately from being a chelating agent as other functions.
  19. Line 322,Please list the relevant instructions on the “health concerns” of fish collagen in the article.

Comments on the Quality of English Language

English quality still needs improvement.

Author Response

In this study, marine-derived collagen and mammalian-derived collagen are compared, and the efficiency and physical and chemical properties of collagen extracted by traditional and new methods are discussed, and the application of collagen is finally expanded. However, there are some several scientific concerns about this manuscript.

Lines18-19 and 24-25, These contents are not reflected in the title, and the title is limited.

Change Title: A review of classical and rising approaches the extraction and utilization of marine collagen

Lines39 and 43-45, "Traditionally, mammalian 38 sources have dominated collagen production" and “Marine-derived collagen, particularly from fish skin, presents a multitude of advantages over its mammalian counterparts. It exhibits lower immunogenicity” lack reference support.

Lines39 References in the text: (Benjakul et al. 2012; Gauza-Włodarczyk et al. 2017).

Line 43 to 45. References for the text: (Sousa et al. 2020; Pal and Suresh 2016)

Lines121-122, Please determine whether the decimal places such as 0.51% and 1.11% are correct. (Lino-Sánchez et al. 2023)

This decimal place is correct.

Figure 9. Yields of collagens extracted from the five batches of scales. (a) Dry weights (in grams) of collagens extracted from the indicated weight of scales. (b) Collagen yields of the five batches calculated as (Dry collagen weight/Dry scales weight) × 100. The mean value of first extractions (A, B, and C) is 0.73% indicated with a blue dashed line, and the mean value of only PSC samples is 0.8675% indicated with a green continuous line.

Lines134-140, Please determine the significance of this reference? This paragraph mainly discusses the traditional acquisition of collagen.

This hypothesis was confirmed by DSC analysis; the denaturation temperatures of the skin and swim bladder were 29.8°C and 32.5°C, respectively. The higher denaturation temperature and hydroxyproline content demonstrated the superior quality of swim bladder collagen. In this sense, (García-Sifuentes et al. 2021), obtained ASC and PSC from discarded fish products. The ASC contained 7% of hydroxyproline, while the PSC contained 6%. The denaturation temperature was 38.27°C and 38.07°C for ASC and PSC, respectively. Although the extraction based on acetic acid showed a higher yield than the pepsin-based protocol (4.4% vs. 2.2%), the quality of the collagen was comparable, considering the hydroxyproline content and denaturation temperature were similar.

This reference explains the characterization of the extracted collagen, hence the importance of the data in this paragraph.

Lines159-161, Please unify the expression of collagen yield. In the article, “45% collagen yield (wet basis)”, “25% of collagen” and “83 and 2.91 mg/g” cannot be compared, and the conclusion cannot be convincing.

The revised texts do not allow data homogenization. It is a bit complicated, in our case we do not want to carry out conversions to avoid an error in the calculations.

Table 1, The content shown in Table 1 is only the collagen yield of different parts of different species under different treatment conditions, so the title cannot be expressed as "Yields of acid-soluble collagen and pepsin-soluble collagen with and without ultrasound assistance".

Change for “Table 1. The collagen yield of different parts of different species under different treatment conditions”

Lines 174-175, Scientists have taken advantage of US technology to improve acid and pepsin-based collagen extraction yields (Table 1)". This sentence is recommended to delete because the contents in the table are already displayed below.

Done. Change text.

Table 1. The collagen yield of different parts of different species under different treatment conditions.

Lines 191-199, This part should better illustrate that ultrasound technology can produce collagen with better thermal stability. Reference recommendations are replaced.

Reference add, Bavisetty SCB, Karnjanapratum S, Dave J, et al (2024) Ultrasonication on Collagen Yield, Physiochemical and Structural Properties from Seabass (Lates Calcarifer) Scales as Affected by Pretreatment and Extraction Conditions. Nat Life Sci Commun 23:e2024003. https://doi.org/10.12982/NLSC.2024.003

Lines 204-206, It is recommended to discuss the impact of frequency, power, amplitude, and time during ultrasonic treatment on collagen extraction quality, or give examples.

Add text.

Ultrasonic treatment is a widely adopted method for collagen extraction due to its efficiency and effectiveness. The quality of collagen extraction is significantly influenced by several parameters, including frequency, power, amplitude, and time. The frequency of ultrasonic waves determines the energy delivered to the collagen matrix. Higher frequencies (typically above 20 kHz) can enhance the extraction process by breaking down collagen fibers more effectively. However, excessively high frequencies may lead to collagen degradation, reducing its quality (Kim et al. 2012).The power of ultrasonic waves is directly related to the intensity of the treatment. Higher power levels can increase the efficiency of collagen extraction by providing more energy to disrupt the collagen matrix. However, excessive power can cause overheating and potential denaturation of collagen, negatively affecting its quality (Sun et al. 2020). Amplitude refers to the displacement of the ultrasonic probe. Higher amplitudes can enhance the cavitation effect, leading to more efficient collagen extraction. However, like power, excessively high amplitudes can damage the collagen structure, resulting in lower quality collagen (Kim et al. 2013).

Lines 216-219, It is recommended to supplement the collagen yield under this method, and summarize and compare the several new collagen extraction methods and collagen yields introduced in the article.

Table 2, The article does not show the role of glycine, proline and hydroxyproline in the extraction of collagen, and the significance of exploring the content of these three amino acids should be supplemented.

Add text.

Fish collagen is a rich source of amino acids, including glycine, proline, and Hydrox-yproline. These amino acids are crucial for the structural integrity and function of colla-gen. Glycine and proline are collagen's most abundant amino acids, making up about one-third of its total amino acid content. It plays a vital role in maintaining the stability and structure of collagen's triple-helix formation. On the other hand, Hydroxyproline is a unique amino acid found exclusively in collagen. The post-translational modification of proline forms it and is crucial for the stability of the collagen triple helix. The content of Hydroxyproline in fish collagen can vary, but it typically accounts for around 6-7% of the total amino acids (García-Sifuentes et al. 2021b). It can be show that there are variations in the concentrations of the amino acids in Table 2.

Line244, No paragraph above mentioned the relevant methods or concepts for hydrolysis and non-hydrolysis of collagen. This paragraph does not connect with the previous article and is recommended to rewrite it.

Change text for:

2.5. Application of various forms of collagen extraction.

Line 245, Please explain the relationship between the extraction method mentioned above and the hydrolysate and the non-hydrolysate.

Add text.

The key difference lies in the degree of hydrolysis: Non-Hydrolysate, minimal hydrolysis, larger collagen molecules, retains native structure. Hydrolysate: Extensive hydrolysis, smaller peptides, enhanced bioavailability and bioactivity. The choice of extraction method depends on the intended application of the collagen. For example, Acid Soluble Collagen (ASC) might be preferred for biomedical applications requiring structural integrity, while PSC hydrolysates are favored for their functional properties in health and beauty products(Jafari et al. 2020).

Lines 249-250, Please add correct references.

Add references.

In native tissues, collagen configuration confers the mechanical resistance of skin and tendons because of the aligned fibrillar arrangement (Balasubramanian et al. 2012).

Line 265, Please explain what "enzymatic pools" is, and no application examples of enzymatic pools are found below.

The collagen is hydrolyzed by enzymes alone or enzymatic pools. Eliminated text “alone or enzymatic pools”. Pools refers to the mixture of several enzymes.

Lines 275-277, This passage does not explain the advantages of using subcritical water to extract peptides compared to enzymatic hydrolysis.

Subcritical Water to Extract (SWE) uses water at high temperatures (100-374°C) and pressures to maintain it in a liquid state. This method is environmentally friendly as it avoids the use of organic solvents (green technology). SWE can break down proteins into peptides more rapidly than enzymatic hydrolysis. The high temperature and pressure conditions accelerate the hydrolysis process (Efficiency). SWE often results in higher yields of peptides compared to enzymatic hydrolysis due to the efficient breakdown of proteins (Álvarez-Viñas et al. 2020).

Lines 281-303, The content is confusing, and it is recommended to reorganize this part of the content, for example, it is recommended to merge antioxidant abilities (DPPH, OH-, ABTS radicals).

Rewrite text.

Collagen peptides, obtained through various methods, have several applications, particularly in the food industry. (Medina-Medrano et al. 2019) explored the antioxidant capacity of collagen in its non-hydrolyzed and hydrolyzed forms. They found that hydrolyzed fractions, obtained through pepsin treatment, trapped free radicals more effectively than non-hydrolyzed ones due to their lower molecular weight. (Zamorano-Apodaca et al. 2020) obtained hydrolyzed collagen from mixed by-products of fish species. Initially, raw fish was treated with 0.5 M acetic acid at a 1:10 solid/liquid ratio for 24 hours at 25°C. The hydrolyzed fractions were then obtained through enzymatic treatment with alcalase at pH 8. The resulting collagen was Type I, with a molecular weight greater than 200 kDa, a yield of 68%, and 5.46% hydrolysis achieved. After enzymatic treatment, five fractions with molecular weights ranging from 1-27 kDa were identified. These fractions exhibited radical scavenging activity for DPPH and OH radicals, comparable to standard Vitamin C. Moreover, the lower molecular weight fractions were richer in soluble protein, making them suitable for functional foods. In another study, (Hernández-Ruiz et al. 2023) tested non-hydrolysate and hydrolysate collagen against E. coli. They identified two hydrolyzed fractions: F1 with peptides of 5-10 kDa, and F2 with peptides of molecular weights lower than 5 kDa. The authors found that the antibacterial activity of collagen peptides was higher than that of non-hydrolysate collagen, likely due to the methionine content, especially in F1. (Fimbres-Romero et al. 2021) investigated the ability of hydrolyzed collagen extracted from jumbo squid and Sierra fish to inhibit ABTS radicals. They used enzymatic extracts obtained from Sierra fish viscera and jumbo squid hepatopancreas. Their findings revealed that low molecular weight peptides (<3 kDa) showed the best antioxidant performance compared to fractions of 3-10 kDa and >10 kDa. Additionally, (Chel-Guerrero et al. 2020) explored the potential of hydrolyzed collagen from Lionfish muscle as chelating agents for Cu2+ and Fe2+. In this study, the hydrolysate was obtained via two routes: the first using pepsin alone, and the second using pepsin followed by pancreatin.

Lines 306-312, This paragraph is not an explanation of the use of hydrolyzed collagen as a chelating agents studied by Chel-Guerrero et al., and it is recommended that this part be discussed separately from being a chelating agent as other functions.

Rewrite text.

The findings revealed that hydrolyzed collagen obtained using the pepsin/pancreatin system exhibited higher metal chelation activity compared to collagen hydrolyzed with pepsin alone. This enhanced activity was attributed to the higher levels of arginine (Arg) and proline (Pro) obtained through the second enzymatic system. The content of imino and amino acids in collagen hydrolysates determines other functionalities. For instance, (Heidari and Rezaei 2022) obtained hydrolyzed collagen from yellowfin tuna using ultrasonication for different durations (0-20 minutes). The samples treated for 15 minutes showed the highest proline and hydroxyproline content. This increase in proline and hydroxyproline content positively impacted the emulsifying and foaming properties, making them useful for food and beverage applications (Pezeshk et al. 2022); (Heidari and Rezaei 2022).

Line 322, Please list the relevant instructions on the “health concerns” of fish collagen in the article.

Add text.

Fish collagen has great potential to replace mammalian collagen without health concerns (digestive issues, allergic reactions, unpleasant taste, and regulation and efficacy).

REFERENCES

Álvarez-Viñas M, Rodríguez-Seoane P, Flórez-Fernández N, et al (2020) Subcritical Water for the Extraction and Hydrolysis of Protein and Other Fractions in Biorefineries from Agro-food Wastes and Algae: a Review. Food Bioprocess Technol 2020 143 14:373–387. https://doi.org/10.1007/S11947-020-02536-4

Bai C, Wei Q, Ren X (2017) Selective Extraction of Collagen Peptides with High Purity from Cod Skins by Deep Eutectic Solvents. ACS Sustain Chem Eng 5:7220–7227. https://doi.org/10.1021/ACSSUSCHEMENG.7B01439/SUPPL_FILE/SC7B01439_SI_001.PDF

Balasubramanian P, Prabhakaran MP, Sireesha M, Ramakrishna S (2012) Collagen in Human Tissues: Structure, Function, and Biomedical Implications from a Tissue Engineering Perspective. Adv Polym Sci 251:173–206. https://doi.org/10.1007/12_2012_176

Benjakul S, Nalinanon S, Shahidi F (2012) Fish Collagen. Food Biochem Food Process Second Ed 365–387. https://doi.org/10.1002/9781118308035.CH20

Chel-Guerrero L, Cua-Aguayo D, Betancur-Ancona D, et al (2020) Antioxidant and chelating activities from Lion fish (Pterois volitans L.) muscle protein hydrolysates produced by in vitro digestion using pepsin and pancreatin. Emirates J Food Agric 32:62–72. https://doi.org/10.9755/EJFA.2020.V32.I1.2060

Fimbres-Romero M de J, Cabrera-Chávez F, Ezquerra-Brauer JM, et al (2021) Utilisation of collagenolytic enzymes from sierra fish (Scomberomorus sierra) and jumbo squid (Dosidicus gigas) viscera to generate bioactive collagen hydrolysates from jumbo squid muscle. J Food Sci Technol 58:2725–2733. https://doi.org/10.1007/S13197-020-04780-0/METRICS

Gauza-WÅ‚odarczyk M, Kubisz L, Mielcarek S, WÅ‚odarczyk D (2017) Comparison of thermal properties of fish collagen and bovine collagen in the temperature range 298–670 K. Mater Sci Eng C 80:468–471. https://doi.org/10.1016/J.MSEC.2017.06.012

Gouarderes S, Ober C, Doumard L, et al (2022) Pulsed Electric Fields Induce Extracellular Matrix Remodeling through Matrix Metalloproteinases Activation and Decreased Collagen Production. J Invest Dermatol 142:1326-1337.e9. https://doi.org/10.1016/J.JID.2021.09.025

Heidari MG, Rezaei M (2022) Extracted pepsin of trout waste and ultrasound-promoted method for green recovery of fish collagen. Sustain Chem Pharm 30:100854. https://doi.org/10.1016/J.SCP.2022.100854

Hernández-Ruiz KL, López-Cervantes J, Sánchez-Machado DI, et al (2023) Collagen peptide fractions from tilapia (Oreochromis aureus Steindachner, 1864) scales: Chemical characterization and biological activity. Food Biosci 53:102658. https://doi.org/10.1016/J.FBIO.2023.102658

Jafari H, Lista A, Siekapen MM, et al (2020) Fish Collagen: Extraction, Characterization, and Applications for Biomaterials Engineering. Polym 2020, Vol 12, Page 2230 12:2230. https://doi.org/10.3390/POLYM12102230

Kim HK, Kim YH, Kim YJ, et al (2012) Effects of ultrasonic treatment on collagen extraction from skins of the sea bass Lateolabrax japonicus. Fish Sci 78:485–490. https://doi.org/10.1007/S12562-012-0472-X/METRICS

Kim HK, Kim YH, Park HJ, Lee NH (2013) Application of ultrasonic treatment to extraction of collagen from the skins of sea bass Lateolabrax japonicus. Fish Sci 79:849–856. https://doi.org/10.1007/S12562-013-0648-Z/METRICS

Lin CC, Wu PT, Chang CW, et al (2020) A single-pulsed electromagnetic field enhances collagen synthesis in tendon cells. Med Eng Phys 77:130–136. https://doi.org/10.1016/j.medengphy.2019.12.001

Lino-Sánchez A, González-Vélez V, Vélez M, Aguilar-Pliego J (2023) Extraction and characterization of type i collagen from scales of Mexican Biajaiba fish. Open Chem 21:. https://doi.org/10.1515/CHEM-2023-0134/MACHINEREADABLECITATION/RIS

Medina-Medrano JR, Quiñones-Muñoz TA, Arce-Ortíz A, et al (2019) Antioxidant Activity of Collagen Extracts Obtained from the Skin and Gills of Oreochromis sp. https://home.liebertpub.com/jmf 22:722–728. https://doi.org/10.1089/JMF.2019.0013

Pezeshk S, Rezaei M, Abdollahi M (2022) Impact of ultrasound on extractability of native collagen from tuna by-product and its ultrastructure and physicochemical attributes. Ultrason Sonochem 89:106129. https://doi.org/10.1016/J.ULTSONCH.2022.106129

Sun M, Wei X, Wang H, et al (2020) Structure Restoration of Thermally Denatured Collagen by Ultrahigh Pressure Treatment. Food Bioprocess Technol 13:367–378. https://doi.org/10.1007/S11947-019-02389-6/METRICS

Zamorano-Apodaca JC, García-Sifuentes CO, Carvajal-Millán E, et al (2020) Biological and functional properties of peptide fractions obtained from collagen hydrolysate derived from mixed by-products of different fish species. Food Chem 331:127350. https://doi.org/10.1016/J.FOODCHEM.2020.127350

Round 2

Reviewer 3 Report

Comments and Suggestions for Authors

This manuscript summarizes the extraction and utilization of marine collagen, which provides theoretical support for the future utilization of marine resources. The whole manuscript is logical and gives a detailed description of the extraction method and processing technology of collagen. However, the whole manuscript lacks diagrams. If possible, it is suggested to add them, which can make a scientific problem more interesting.

In addition, I would like to ask the author what is the focus of Marine collagen research in the future, especially in terms of new applications and extraction technologies. Hope the author can answer, thank you!

Author Response

Reviewer

This manuscript summarizes the extraction and utilization of marine collagen, which provides theoretical support for the future utilization of marine resources. The whole manuscript is logical and gives a detailed description of the extraction method and processing technology of collagen. However, the whole manuscript lacks diagrams. If possible, it is suggested to add them, which can make a scientific problem more interesting.

Dear reviewer, thank you very much for your positive contributions to the submitted work. A figure has been added with diagrams of the classic methods and emerging approaches to collagen extraction.

Figure 1. Graphic representation of collagen extraction methods. a) Extraction of marine collagen whit classical methods. b) Extraction of marine collagen using new methods.

In addition, I would like to ask the author what is the focus of Marine collagen research in the future, especially in terms of new applications and extraction technologies. Hope the author can answer, thank you!

This is a verbatim quote from non-classical methods. “The application of these novel technologies reduces the substantial quantities of chemicals used in collagen extraction in addition to reducing the extraction time without jeopardizing the collagen yield and functionality”.(Vate and Abdollahi 2025)

The future of extraction methods lies in finding a fast, efficient, and easy-to-use method that saves time in purification and ensures the quality of the desired product. Microwave and ultrasonic extraction methods offer minimal manipulation by the analyst, excellent extraction times ranging from 15 to 30 minutes, and a variety of power levels in modern equipment.(Kendler et al. 2023)

 (Gaikwad and Kim 2024)advocate for research into various combination strategies, as relying solely on individual polymers does not meet the diverse needs of different applications. To produce suitable collagen from fish sources, it is essential to explore innovative modifications in the physical, chemical, and enzymatic properties of fish collagen. This approach can uncover potential marine sources that address current challenges and contribute to the development of versatile fish collagen-based composite materials across various industries.

References

Gaikwad, Sunita, and Mi Jeong Kim. 2024. “Fish By-Product Collagen Extraction Using Different Methods and Their Application.” Marine Drugs 2024, Vol. 22, Page 60 22(2): 60. doi:10.3390/MD22020060.

Kendler, Sophie, Sine Marie Moen Kobbenes, Anita Nordeng Jakobsen, Kirill Mukhatov, and Jørgen Lerfall. 2023. “The Application of Microwave and Ultrasound Technologies for Extracting Collagen from European Plaice By-Products.” Frontiers in Sustainable Food Systems 7: 1257635. doi:10.3389/FSUFS.2023.1257635/BIBTEX.

Vate, Naveen Kumar, and Mehdi Abdollahi. 2025. “Marine Collagens and Novel Insights in Their Sustainable Extraction.” : 253–71. doi:10.1007/978-981-96-1253-6_11.
